**Data Availability Statement:** The data used in this analysis are third party data not owned by the authors and therefore cannot be shared publicly.

# Risk factors and genotype distribution of hepatitis C virus in Georgia: A nationwide population-based survey

Davit Baliashvili[1,2]*, Francisco Averhoff[3], Ana Kasradze[1], Stephanie J. Salyer[4], Giorgi Kuchukhidze[1], Amiran Gamkrelidze[1], Paata Imnadze[1], Maia Alkhazashvili[1], Gvantsa Chanturia[1], Nazibrola Chitadze[1], Roena Sukhiashvili[1], Curtis Blanton[4], Jan Drobeniuc[3], Juliette Morgan[4,5], Liesl M. Hagan[3]

**1** National Center for Disease Control and Public Health, Tbilisi, Georgia, **2** Department of Epidemiology, Emory University Rollins School of Public Health, Atlanta, Georgia, United States of America, **3** Division of Viral Hepatitis, Centers for Disease Control and Prevention, Atlanta, Georgia, United States of America, **4** Division of Global Health Protection, Centers for Disease Control and Prevention, Atlanta, Georgia, United States of America, **5** Global Disease Detection – South Caucasus Regional Center, Centers for Disease Control and Prevention, Tbilisi, Georgia

* davit.baliashvili@emory.edu, dato.baliashvili@gmail.com

## Abstract

In preparation for the National Hepatitis C Elimination Program in the country of Georgia, a nationwide household-based hepatitis C virus (HCV) seroprevalence survey was conducted in 2015. Data were used to estimate HCV genotype distribution and better understand potential sex-specific risk factors that contribute to HCV transmission. HCV genotype distribution by sex and reported risk factors were calculated. We used explanatory logistic regression models stratified by sex to identify behavioral and healthcare-related risk factors for HCV seropositivity, and predictive logistic regression models to identify additional variables that could help predict the presence of infection. Factors associated with HCV seropositivity in explanatory models included, among males, history of injection drug use (IDU) (aOR = 22.4, 95% CI = 12.7, 39.8) and receiving a blood transfusion (aOR = 3.6, 95% CI = 1.4, 8.8), and among females, history of receiving a blood transfusion (aOR = 4.0, 95% CI 2.1, 7.7), kidney dialysis (aOR = 7.3 95% CI 1.5, 35.3) and surgery (aOR = 1.9, 95% CI 1.1, 3.2). The male-specific predictive model additionally identified age, urban residence, and history of incarceration as factors predictive of seropositivity and were used to create a male-specific exposure index (Area under the curve [AUC] = 0.84). The female-specific predictive model had insufficient discriminatory performance to support creating an exposure index (AUC = 0.61). The most prevalent HCV genotype (GT) nationally was GT1b (40.5%), followed by GT3 (34.7%) and GT2 (23.6%). Risk factors for HCV seropositivity and distribution of HCV genotypes in Georgia vary substantially by sex. The HCV exposure index developed for males could be used to inform targeted testing programs.

The requests for the data sharing should be sent to the National Center for Disease Control and Public Health of Georgia via email: pr.ncdc@ncdc.ge. We confirm that others would be able to access these data in the same manner as the authors. The authors did not have any special access privileges that others would not have.

**Funding:** This study analyzed data from a serosurvey originally funded by the government of Georgia and the USCDC and previously published (Hagan et.al., 2019). This work was supported in part by the NIH Fogarty International Center Global Infectious Diseases grant D43TW007124 (to DB). There was no additional external funding received for this study.

**Competing interests:** The authors have declared that no competing interests exist.

## Introduction

The World Health Organization (WHO) estimates that in 2015, 71 million people globally were living with hepatitis C virus (HCV) infection and 400,000 died as a consequence of HCV infection [1, 2]. HCV ribonucleic acid (RNA) can be detected in blood and body fluids including saliva, tears, and semen, but transmission occurs primarily via infected blood or blood-derived body fluids [3–8]. The most common mode of transmission in industrialized countries is sharing needles for injection drug use (IDU) [4, 9, 10]. However, in low and middle-income countries the leading factors contributing to transmission are nosocomial exposures resulting from poor infection control practices and contaminated blood transfusions [4, 9, 11, 12]. Community exposures such as barbering and tattooing have also been reported as risk factors for HCV infection in some countries [13–15]. Perinatal transmission from mother to child can occur, and sexual transmission, primarily among men who have sex with men, has been documented [16–18].

HCV has wide genetic heterogeneity, with seven major genotypes (GT) and 67 subtypes [19]. The most common genotype globally is GT1, which accounts for 44–46% of all HCV infections, followed by GT3 (22–25%) and GT4 (13–15%) [2, 20]. Prevalence of each genotype varies geographically, as well as by different population subgroups. In Central and Eastern Europe, GT1 accounts for more than 60% of all HCV infections, and GT3 is most prevalent among people who inject drugs [2, 21–23].

The country of Georgia is a middle-income Eastern European country with a high burden of hepatitis C. In a nationally representative seroprevalence survey conducted in 2015, 7.7% of the general population tested positive for HCV antibody (anti-HCV), indicating a past or current HCV infection, and 5.4% were living with chronic HCV infection [24]. In 2015, Georgia launched a nationwide hepatitis C elimination program, aiming to reduce the prevalence of HCV infection by 90% through universal access to screening, care, and treatment [25–27].

In our previous analysis of data from the 2015 seroprevalence survey [24], data from males and females were analyzed together, and history of IDU and blood transfusion were the only exposures independently associated with HCV seropositivity. However, because only half of seropositive survey participants reported one of these risk factors, Georgian Ministry of Health officials concluded that a targeted HCV testing strategy based on acknowledged IDU and blood transfusion history alone would not identify the large proportion of individuals who did not have or disclose these risk factors. Therefore, we analyzed the 2015 seroprevalence survey data to identify sex-specific risk factors for HCV infection and the HCV genotype distribution in Georgia, and to develop a prediction tool to better inform HCV testing strategies.

## Materials and methods

### Study population and data collection

Our analysis uses data from a nationally representative, household-based seroprevalence survey conducted in Georgia in 2015. The survey provided an estimate of national hepatitis C prevalence and risk factors associated with infection. The survey used a stratified, multi-stage cluster design with a target sample size of 7,000 adults aged ≥18 years. Data were collected on socio-demographic characteristics, medical history, behavioral exposures, and potential hepatitis C risk factors. Phlebotomists collected a 10 mL blood sample from each participant. Further details regarding sample design and data collection have been described previously [24].

### Laboratory methodology

Laboratory procedures from the seroprevalence survey have been previously described [24]. Briefly, blood samples were tested for anti-HCV antibodies using an enzyme-immunoassay,

anti-HCV positive samples were then tested for HCV RNA to determine active infection (Sacace™ HCV Real-TM Qual, Sacace Biotechnologies, Srl, Italy) and RNA-positive samples were tested for HCV genotype. Genotyping was performed using commercial kit—HCV Real-TM Genotype from Sacace. This Real Time PCR Kit was dedicated for qualitative detection and differentiation of hepatitis C virus (HCV) genotypes 1a, 1b, 2, 3, 4. Analytical sensitivity provided in the instruction was 500 IU/ml. Laboratory staff from the US Centers for Disease Control and Prevention (CDC) monitored protocols and processes for quality assurance and quality control.

## Statistical analysis

Statistical analyses for this study were performed using SAS 9.4 (Cary, North Carolina, USA). Seroprevalence survey data were weighted based on probability of selection at cluster, house-hold, and individual levels using 2014 census data, and analyses used complex survey proce-dures accounting for stratification, clustering, and unequal sample weights (SAS procedures SURVEYFREQ and SURVEYLOGISTIC). The HCV genotype distribution was calculated for the overall population, as well as by sex, age and reported risk factors. Weighted anti-HCV prevalence estimates, as well as unadjusted odds ratios and 95% confidence intervals (CI) were calculated for males and females separately. Descriptive analysis of the distribution of self-reported risk factors was calculated separately among males and females. History of IDU was included in this descriptive analysis for both males and females because it is a well-known risk factor for HCV infection regardless of sex, but it was not retained in the female-specific multi-variable analysis due to small cell size.

We conducted two separate multivariable analyses using logistic regression models—explanatory and predictive regression models stratified by sex. In the explanatory logistic regression models, we estimated sex-specific associations between HCV seropositivity and self-reported behavioral and healthcare-related exposures that could be causally associated with HCV infection. Exposure variables were included in the regression models based on exist-ing literature and results of unadjusted analyses. Potential confounders were identified by reviewing the existing literature and using directed acyclic graphs (DAGs) [28]. Collinearity was assessed in the final sex-specific models using condition indices and variance decomposi-tion proportions [29]. Adjusted odds ratios and 95% CIs are presented.

In the original seroprevalence survey analysis, 46.7% of anti-HCV positive participants reported neither of the two risk factors identified in regression models as independent risk fac-tors for hepatitis C (history of IDU or blood transfusion) [24]. To create a more sensitive screening tool to inform targeted HCV testing efforts, we built sex-specific predictive logistic regression models to identify additional variables that can predict seropositivity, even if they are not causally associated with the infection. Predictive regression models included risk fac-tors identified in the explanatory regression models described above, as well as additional behavioral and socio-demographic variables associated with seropositivity in the unadjusted analysis (Table 1). Age variable in predictive models was dichotomized with cut-point selected based on age and sex-specific prevalence trends from the previously reported analysis [24]. Final variable selection was performed manually, using a 60% subset of the data from the 2015 seroprevalence survey (training set). We removed variables from the initial models if they did not provide stable estimates (e.g. due to low numbers). Next, we removed variables that were not significantly associated with seropositivity in the predictive model (significance level $\alpha$ = .05) and if their removal did not change the model's discriminatory performance, measured by area under the receiver operating characteristic curves (AUC). Final predictive models were validated using the remaining 40% of serosurvey data (validation set).

**Table 1. Descriptive statistics and results of unadjusted and adjusted analyses of anti-HCV risk factors, stratified by sex, Georgia HCV serosurvey, 2015.**

| Characteristic | Males | | | | | Females | | | | |
|---|---|---|---|---|---|---|---|---|---|---|
| | Total (n = 2,339)† | Anti-HCV positive (n = 288) | | Unadjusted OR (95% CI) | aOR‡ (95% CI) | Total (n = 3,671)† | Anti-HCV positive (n = 145) | | Unadjusted OR (95% CI) | aOR‡ (95% CI) |
| | | n | Weighted % (95% CI) | | | | n | Weighted % (95% CI) | | |
| **Geography** | | | | | | | | | | |
| Urban | 1,249 | 208 | 15.8 (12.4, 19.1) | 2.3 (1.6, 3.3) | | 1,906 | 82 | 4.2 (2.7, 5.6) | 1.2 (0.8, 2.0) | |
| Rural | 1,090 | 80 | 7.7 (5.8, 9.6) | 1 | | 1,765 | 63 | 3.4 (2.3, 4.4) | 1 | |
| **Ever injected drugs** | | | | | | | | | | |
| Yes | 202 | 148 | 67.0 (57.1, 77.0) | 28.8 (17.5, 47.6) | 22.4 (12.7, 39.8) | 3 | 2 | 45.2 (0.0, 100.0) | 21.2 (1.7, 271.1)¶ | |
| No | 2,123 | 140 | 6.6 (5.1, 8.1) | 1 | 1 | 3,639 | 143 | 3.8 (2.9, 4.6) | 1 | |
| **Ever incarcerated** | | | | | | | | | | |
| Yes | 224 | 96 | 43.2 (33.5, 53.0) | 8.0 (5.2, 12.2) | | 12 | 2 | 7.9 (0.0, 20.1) | 2.2 (0.4, 11.8) | |
| No | 2,109 | 192 | 8.7 (7.0, 10.4) | 1 | | 3,648 | 143 | 3.8 (2.9, 4.7) | 1 | |
| **Have any tattoos** | | | | | | | | | | |
| Yes | 586 | 103 | 17.3 (12.5, 22.0) | 1.8 (1.2, 2.6) | | 40 | 1 | 1.1 (0.0, 3.2) | 0.3 (0.04, 2.05) | |
| No | 1,749 | 185 | 10.4 (8.3, 12.5) | 1 | | 3,623 | 144 | 3.9 (2.9, 4.8) | 1 | |
| **Have any piercings** | | | | | | | | | | |
| Yes | 5 | 0 | 0 | - | | 2,708 | 98 | 3.9 (2.7, 5.0) | 1.1 (0.6, 1.8) | |
| No | 2,330 | 288 | 12.2 (10.1, 14.3) | - | | 954 | 47 | 3.7 (2.2, 5.2) | 1 | |
| **Ever received a blood transfusion** | | | | | | | | | | |
| Yes | 156 | 37 | 30.8 (19.1, 42.4) | 3.7 (2.2, 6.2) | 3.6 (1.4, 8.8) | 291 | 32 | 14.0 (6.8, 21.2) | 5.3 (2.8, 10.1) | 4.0 (2.1, 7.7) |
| No | 2,180 | 251 | 10.8 (9.0, 12.6) | | 1 | 3,374 | 113 | 3.0 (2.3, 3.7) | 1 | 1 |
| **Ever received kidney dialysis** | | | | | | | | | | |
| Yes | 6 | 1 | 14.3 (0.0, 41.8) | 1.2 (0.1, 11.6) | | 11 | 2 | 37.8 (0.0, 78.6) | 15.8 (2.8, 88.1) | 7.3 (1.5, 35.3) |
| No | 2,327 | 287 | 12.1 (10.1, 14.2) | 1 | | 3,645 | 143 | 3.7 (2.8, 4.6) | 1 | |
| **Frequency of dental cleanings** | | | | | | | | | | |
| Twice per year | 68 | 9 | 26.6 (9.4, 43.8) | 2.9 (1.2, 7.0) | | 125 | 6 | 4.0 (0.1, 7.9) | 1.0 (0.4, 2.5) | |
| Once per year | 170 | 22 | 13.2 (4.8, 21.7) | 1.2 (0.6, 2.6) | | 308 | 5 | 2.1 (0.0, 4.9) | 0.5 (0.1, 1.8) | |
| Less frequently than once per year | 387 | 61 | 11.5 (7.5, 15.6) | 1.0 (0.7, 1.6) | | 721 | 23 | 3.6 (1.4, 5.8) | 0.9 (0.4, 1.7) | |
| Never | 1,692 | 194 | 11.2 (9.2, 13.2) | 1 | | 2,481 | 110 | 4.1 (3.1, 5.2) | 1 | |
| **Ever had surgery** | | | | | | | | | | |
| Yes | 1,150 | 146 | 13.0 (9.7, 16.3) | 1.2 (0.8, 1.7) | | 2,358 | 110 | 4.8 (3.5, 6.1) | 2.4 (1.4, 3.9) | 1.9 (1.1, 3.2) |
| No | 1,184 | 142 | 11.2 (8.7, 13.8) | 1 | | 1,298 | 34 | 2.1 (1.2, 3.0) | 1 | 1 |
| **Ever had a manicure or pedicure in a salon** | | | | | | | | | | |
| Yes | 37 | 6 | 19.7 (2.4, 36.9) | 1.8 (0.6, 5.3) | | 916 | 40 | 4.1 (2.0, 6.2) | 1.1 (0.6, 2.1) | |
| No | 2,299 | 282 | 11.9 (9.9, 13.9) | 1 | | 2,749 | 105 | 3.7 (2.7, 4.8) | 1 | |

*(Continued)*

**Table 1.** (Continued)

| Characteristic | Males | | | | | Females | | | |
| --- | --- | --- | --- | --- | --- | --- | --- | --- | --- |
| | Total (n = 2,339)† | Anti-HCV positive (n = 288) | | Unadjusted OR (95% CI) | aOR‡ (95% CI) | Total (n = 3,671)† | Anti-HCV positive (n = 145) | | Unadjusted OR (95% CI) | aOR‡ (95% CI) |
| | | n | Weighted % (95% CI) | | | | n | Weighted % (95% CI) | | |
| **Typically shave in barber or salon§** | | | | | | | | | | |
| Yes | 319 | 48 | 14.4 (8.6, 20.3) | 1.3 (0.8, 2.1) | | - | - | - | | |
| No | 2,017 | 240 | 11.8 (9.6, 13.9) | 1 | | - | - | - | | |
| **Number of lifetime sex partners** | | | | | | | | | | |
| >2 | 968 | 126 | 11.8 (8.7, 14.9) | 3.1 (1.4, 6.7) | | 23 | 3 | 16.3 (0.0, 35.3) | 5.0 (1.3, 20.0) | |
| ≤2 | 400 | 16 | 4.1 (1.4, 6.8) | 1 | | 3,620 | 141 | 3.7 (0.4, 4.6) | 1 | |

Abbreviations: HCV = Hepatitis C virus, CI = Confidence Interval, aOR = Adjusted Odds Ratio.

† Individual cells under each variable might not sum up to total due to the missing values not included in the table.

‡ Adjusted models included all variables reported in this column, in addition to control variables (age, geography (urban vs rural) and history of incarceration).

§ Only men were asked this question.

⁋ IDU was not included in the multivariable model for females due to the small number of females reporting IDU.

We used the variables in the final male-specific predictive model to create a male-specific exposure index that can be applied to individuals to predict their likelihood of being infected with HCV. To create the male-specific exposure index, we assigned a risk score to each variable in the final predictive model using the following formula: parameter estimate from final predictive model multiplied by 5 and rounded to the nearest whole number (Fig 1). To test the discriminatory performance of the exposure index, we then assigned each male serosurvey participant a cumulative risk score based on reported risk factors and demographics, and ran an additional male-specific logistic regression model with the risk score as the only predictor variable and HCV infection status as the outcome variable. We also calculated the distribution of risk scores across male seroprevalence survey participants. The discriminatory performance of the female-specific predictive model was not high enough to create a female-specific exposure index.

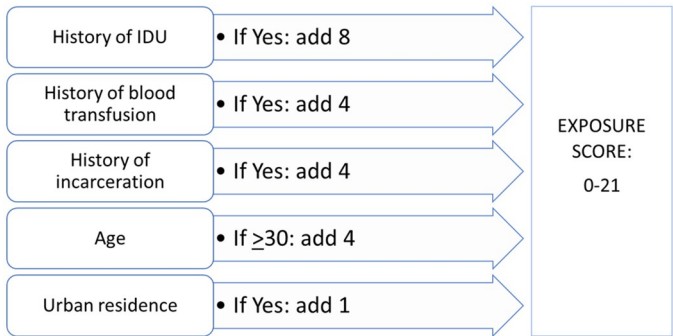

**Fig 1. Calculation of HCV exposure score for males.** *Note*: Exposure score points were assigned to each variable in the final predictive model using the following formula: parameter estimate from final predictive model multiplied by 5 and rounded to the nearest whole number. *Abbreviations*: HCV = hepatitis C virus, IDU = Injection drug use.

### Ethics

Ethical approval was obtained from the Georgian National Center for Disease Control and Public Health Institutional Review Board. CDC's Human Subjects Research Office determined this survey to be a routine public health activity for public health surveillance, therefore judged to not involve human subjects research and the need for consent was waived. The study did not involve minors.

## Results

### Risk factors for anti-HCV positivity

Descriptive analysis of the study population and weighted estimates of nationwide hepatitis C prevalence were reported previously [24]. To summarize briefly, the final sample with available HCV antibody testing results (n = 6010) included a total of 2,339 (39%) males, with a weighted national HCV seroprevalence of 12.1% (95% CI: 10.2, 14.3; n = 288), and 3,671 (61%) females, with a weighted national HCV seroprevalence of 3.8% (95% CI: 3.0, 4.9; n = 145) [24]. A total of 311 survey participants were found HCV-RNA positive, 218 of them among males (weighted prevalence 9.0%) and 93 among females (weighted prevalence 2.2%).

In our current analysis, prevalence of anti-HCV among males was highest among those who reported history of IDU (67.0%), history of incarceration (43.2%), history of receiving a blood transfusion (30.8%), dental cleaning twice per year (26.6%), having any tattoo (17.3%), and those living in urban areas (15.8%) (Table 1). Among females, anti-HCV prevalence was highest among those reporting history of receiving a blood transfusion (14.0%) and history of surgery (4.8%) (Table 1). Due to the small numbers of females reporting history of IDU (n = 3), history of incarceration (n = 12), tattoos (n = 40), ever receiving kidney dialysis (n = 11), or having more than two lifetime sex partners (n = 23), and even smaller numbers of seropositive females with these reported risk factors, it was not possible to reliably estimate prevalence of anti-HCV among women reporting these risk factors.

In explanatory multivariable logistic regression models stratified by sex and adjusted for age, incarceration history and urban geography, two exposures were independent risk factors for anti-HCV positivity among males: history of IDU (aOR = 22.4, 95% CI: 12.7, 39.8) and history of receiving a blood transfusion (aOR = 3.6, 95% CI: 1.4, 8.8), similar to the results from the overall population. Among females, independent risk factors for anti-HCV positivity were history of receiving a blood transfusion (aOR = 4.0, 95% CI: 2.1, 7.7) and history of surgery (aOR = 1.9, 95% CI: 1.1, 3.2). History of receiving kidney dialysis was also strongly associated with anti-HCV positivity among females (aOR = 7.3, 95% CI: 1.5, 35.3); however, due to the small number of females reporting this risk factor (n = 11), the estimate is imprecise. History of IDU was not included in the female-specific model due to insufficient statistical power (only 3 females reported history of IDU) (Table 1).

Among anti-HCV positive participants, 61.6% of males and 84.3% of females reported at least one of the risk factors found to be independently associated with seropositivity in the sex-specific explanatory multivariable models. Among anti-HCV positive males, 50.9% reported history of IDU, and 16.9% reported history of blood transfusion, including 6.2% who reported both of those risk factors; among anti-HCV positive females, 2.4% reported history of IDU, 27.6% reported receiving a blood transfusion, and 80% reported history of surgery, including 25.7% who reported a combination of these factors (Fig 2).

Self-reported risk factors varied by age in both males and females. History of IDU was most commonly reported by anti-HCV positive males in age categories below 50 (55.2%, 54.7% and 64.9%, in age groups 18–29, 30–39 and 40–49, respectively. S1 Fig in S1 Appendix). The

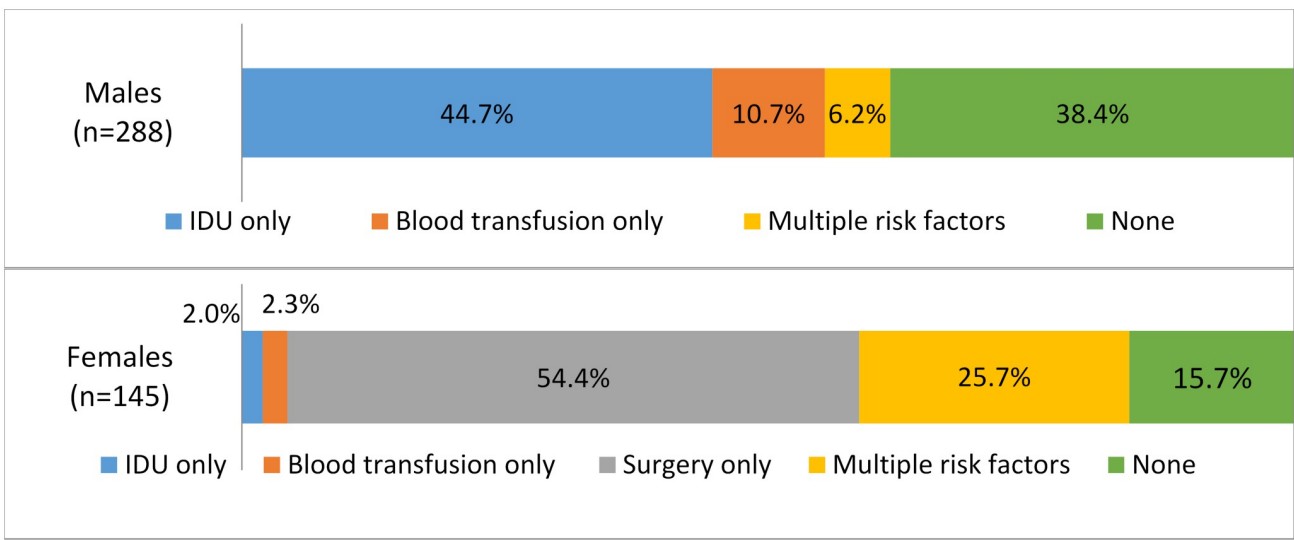

**Fig 2. Percent of anti-HCV+ participants self-reporting hepatitis C risk factors, stratified by sex, Georgia HCV serosurvey, 2015.** *Note*: History of surgery was only explored as risk factor of interest among females. History of IDU was not included in the female-specific multivariable models of risk factor analysis due to low numbers, but is retained in descriptive analyses since it is a known risk factor regardless of sex. *Abbreviations*: HCV = hepatitis C virus, Anti-HCV = antibodies against hepatitis C virus, IDU = Injection drug use.

proportion of anti-HCV positive males who did not report one of the male-specific risk factors for HCV seropositivity (history of IDU or blood transfusion) was higher in age groups above 50, comprising 45.5% among males aged 50–59 and 55.6% among males aged ≥60. Among anti-HCV positive females, the highest proportion of participants reporting no history of any risk factors for HCV seropositivity (IDU, blood transfusion and surgery), were aged ≥60 (21%). However, in the 50–59 age group, only 6% of females reported none of the three major risk factors. (S2 Fig in S1 Appendix).

### Genotype distribution

HCV genotype testing was performed on samples from the 310 RNA-positive serosurvey participants. The most prevalent genotype nationally was GT1b (40.5%), followed by GT3 (34.7%), GT2 (23.6%) and GT1a (0.6%). Five participants (0.7%) had indeterminate genotype results and were removed from further analysis (Table 2).

Genotype distribution varied by sex and reported risk factors, with GT3 the most common genotype among males (39.8%) and among participants of both sexes reporting history of IDU (39.4%). GT1b was most prevalent among females (62.5%) and among participants of both sexes reporting history of receiving a blood transfusion (46.2%). Among participants reporting no sex-specific risk factors for seropositivity, GT3 was the most common among males (41.9%), and GT1b was most common among females (93.6%). Genotype distribution also varied by age group, with GT1b accounting for almost three-quarters of all RNA-positive participants above age 60, and GT3 accounting for a larger percentage of infections in younger age groups (Table 2).

### Predictive model and exposure index

The final predictive model used to inform the male-specific exposure index included history of IDU, history of blood transfusion, history of incarceration, urban vs rural residence, and a

**Table 2. HCV genotype distribution by sex and reported risk factors, Georgia HCV serosurvey, 2015.**

| Population group | Weighted percentage of each genotype† | | | |
|---|---|---|---|---|
| | GT1a (n = 3) | GT1b (n = 132) | GT2 (n = 73) | GT3 (n = 97) |
| **Overall (N = 310) ‡** | **0.6%** | **40.5%** | **23.6%** | **34.7%** |
| 18–29 (n = 13) | 0.0% | 48.1% | 1.4% | 50.4% |
| 30–39 (n = 72) | 1.7% | 37.4% | 19.6% | 41.3% |
| 40–49 (n = 96) | 0.0% | 28.9% | 23.3% | 47.7% |
| 50–59 (n = 61) | 0.9% | 38.9% | 37.1% | 23.0% |
| 60+ (n = 63) | 0.0% | 74.9% | 23.6% | 1.5% |
| **Males (n = 216)** | **0.7%** | **34.9%** | **24.5%** | **39.8%** |
| Reported history of IDU (n = 110) | 1.0% | 28.9% | 30.2% | 39.9% |
| Reported history of blood transfusion (n = 27) | 0.0% | 45.2% | 8.6% | 46.2% |
| No reported risk factors (n = 88) § | 0.5% | 37.4% | 20.2% | 41.9% |
| 18–29 (n = 9) | 0.0% | 51.3% | 2.0% | 46.7% |
| 30–39 (n = 58) | 2.0% | 33.2% | 23.2% | 41.6% |
| 40–49 (n = 82) | 0.0% | 25.1% | 24.4% | 50.5% |
| 50–59 (n = 40) | 1.4% | 43.0% | 24.1% | 31.5% |
| 60+ (n = 27) | 0.0% | 60.6% | 38.1% | 1.2% |
| **Females (n = 89)** | **0.0%** | **62.5%** | **20.9%** | **16.6%** |
| Reported history of Blood transfusion (n = 18) | 0.0% | 49.1% | 43.8% | 7.0% |
| Reported history of surgery (n = 66) | 0.0% | 54.0% | 25.3% | 20.7% |
| No reported risk factors (n = 18)⁋ | 0.0% | 93.6% | 0.0% | 6.4% |
| 18–29 (n = 4) | 0.0% | 39.7% | 0.0% | 60.3% |
| 30–39 (n = 14) | 0.0% | 60.2% | 0.0% | 39.8% |
| 40–49 (n = 14) | 0.0% | 65.8% | 13.3% | 21.0% |
| 50–59 (n = 21) | 0.0% | 30.5% | 63.6% | 5.9% |
| 60+ (n = 36) | 0.0% | 92.9% | 5.3% | 1.8% |

Abbreviations: HCV = Hepatitis C virus, GT = genotype, IDU = injection drug use.

† Percentages within each category might not sum up to 100% due to the rounding error.

‡ N = 310 includes 5 participants with indeterminate genotype result, but they are excluded from further calculations of group-specific percentages;

§ Did not report history of IDU or blood transfusion;

⁋ Did not report history of IDU, surgery or blood transfusion. Only two females reported history of IDU, one with GT1b and another with GT3.

binary age variable dichotomized at 30 years (Table 3). The model was built using randomly selected 60% of the male subset of the serosurvey data (n = 1,490) and validated on the remaining 40% (n = 938). This model showed high discriminatory performance, with AUC = 0.84 in the training dataset (S3a Fig in S1 Appendix) and AUC = 0.85 in the validation dataset. Adding other exposure variables such as ever having a tattoo, piercing, surgery, and typically being

**Table 3. Final HCV predictive model for males, parameter estimates and score assigned.**

| Variable | Parameter Estimate | Standard error | P-value | Score assigned in exposure index |
|---|---|---|---|---|
| Ever received blood transfusion | 0.75 | 0.22 | < .01 | 4 |
| Ever receiving injection drug use | 1.55 | 0.19 | < .01 | 8 |
| Urban residence | 0.29 | 0.14 | .03 | 1 |
| Ever incarcerated | 0.85 | 0.18 | < .01 | 4 |
| Age >30 | 0.84 | 0.36 | .02 | 4 |

shaved in a barber shop or beauty salon did not improve the discriminatory performance of the model.

The male-specific exposure index included the same variables as the final predictive model. When the index was applied to the serosurvey data, male participants' risk scores ranged from 0 to 21 (Fig 1). Among male participants overall, risk scores clustered toward the low end of the distribution. HCV seroprevalence increased by exposure score and ranged from 1.1% among males with an exposure score of 0 or 1 and reached 100% among males with the highest score (Table 4). In the male-only logistic regression model with the exposure index as the single predictor variable, the index showed high discriminatory performance, with AUC = 0.84, matching the results from the predictive models.

The female-specific predictive model was built using 60% of the female subset of the serosurvey data (n = 2368). We were unable to identify any other variables that could help predict HCV seropositivity in addition to significant risk factors from the explanatory model (history of blood transfusion and surgery). The final model that included only these two risk factor variables had low discriminatory performance (AUC = 0.61). The inclusion of other variables (e.g., history of dialysis, urban vs. rural geography and age) did not increase the model's discriminatory performance substantially, with maximum AUC of 0.65 (S3b Fig in S1 Appendix). The discriminatory performance of the female model was insufficient to validate it and create a meaningful exposure index for females.

## Discussion

In this analysis of Georgia's first nationwide HCV seroprevalence survey, we found that HCV transmission among males is likely driven by IDU, while blood transfusion, history of surgery and/or other unidentified risk factors account for a larger proportion of infections among females. We also found that HCV genotype distribution in Georgia varies by sex, age, and self-reported risk factors. In the overall population, genotypes 1b and 3 account for 40.5% and 34.7% of chronic HCV infections, respectively. GT1b was more common among females, persons more likely infected via blood transfusion and persons over the age of 50, while GT3 was

**Table 4. Proportions of anti-HCV positive participants in each of the exposure score categories among males, Georgia HCV serosurvey, 2015.**

| Exposure score† | Total Number | # Anti-HCV positive (weighted %) |
|:---:|:---:|:---:|
| 0 | 153 | 1 (1.1) |
| 1 | 245 | 4 (1.1) |
| 4 | 778 | 27 (3.1) |
| 5 | 725 | 67 (7.7) |
| 8 | 107 | 20 (25.0) |
| 9 | 131 | 24 (24.3) |
| 12 | 30 | 15 (44.4) |
| 13 | 81 | 52 (54.4) |
| 16 | 20 | 16 (83.1) |
| 17 | 62 | 56 (89.0) |
| 20 | 2 | 1 (71.7) |
| 21 | 5 | 5 (100.0) |
| **Total** | **2,339** | **288** |

† Exposure scores with 0 participants are not included in the table.

Abbreviations: HCV = Hepatitis C virus, Anti-HCV = antibodies against hepatitis C virus.

more common among males, persons more likely infected through IDU, and younger age groups.

These sex- and age-based differences in genotype, combined with differences in our sex-specific predictive models, suggest that hepatitis C risk factors may substantially differ by sex and age. The strong association between history of injection drug use and anti-HCV positivity in males, combined with the high prevalence of reported IDU among younger males, highlights the importance of targeting hepatitis C prevention and testing programs to people who inject drugs. Receiving a blood transfusion was strongly associated with having anti-HCV antibodies in both males and females, suggesting the need to improve quality control mechanisms in the national blood safety program. Even though universal HCV antibody screening of blood donations in Georgia started in 1997, our previous analysis found that receiving a blood transfusion after 1997 was still associated with high anti-HCV prevalence, suggesting the need for further improvements in blood safety [24]. However, a recent analysis of blood transfusion programs in Georgia showed positive trends in blood safety since 2015, suggestive of collateral benefit from a national hepatitis C elimination program [30].

History of surgery as a single risk factor or in combination with other factors was reported by a larger proportion of seropositive females (80%) than males (53%) and was associated with anti-HCV positivity only among females. Additional data would be needed to determine the reason for this difference. One dynamic that should be explored is potential exposure to HCV during childbirth by caesarean section (C-section), through an associated blood transfusion that the patient may not recall. The proportion of births involving C-section in Georgia increased markedly during the past several decades, from 3.8% of all births in 1990 to 36.7% in 2012 [31]. Underreporting of blood transfusions during C-section due to incomplete recall could potentially account for the elevated risk associated with surgery only present among females.

Sixteen percent of anti-HCV positive females and 38% of anti-HCV positive males did not report history of any of the sex-specific risk factors found to be independently associated with HCV seropositivity. However, the sex-specific genotype distribution among anti-HCV positive participants who did not report risk factors was similar to that among participants who did report risk factors, indicating that participants may have either chosen not to disclose stigmatizing risk factors (such as IDU), and/or were unable to recall risk-associated events that occurred earlier in life. Underreporting is particularly likely among people who inject drugs, due to historically strict enforcement of laws against drug use in Georgia that were still in place at the time of survey fieldwork [32]. The finding that the proportion of anti-HCV positive males and females who did not report sex-specific risk factors was higher in older age categories supports potential recall bias. It is also possible that other healthcare and community exposures, such as dental procedures, tattoos and piercings could contribute to HCV transmission and we either did not include them in the survey or did not have enough statistical power to identify them as significant risk factors (Table 1).

This study was the first nationwide HCV seroprevalence survey in Georgia, making it challenging to observe the temporal trends in HCV genotype distribution or risk factor profiles in Georgia. However, our findings are mostly comparable to the previous study in the capital city Tbilisi, conducted in 2001–2002. The previous study also found GT1b to be the most common genotype (59%), followed by the GT3 (27%) [33]. In terms of the risk factor distribution, previous study reported much higher proportion of HCV seropositive individuals reporting history of IDU (85%) [34]. This difference could be explained by the fact that IDU behavior is mostly concentrated in the urban area, where the previous study was based.

Our findings have several important implications for the Georgian hepatitis C elimination program, as well as for other countries aiming to scale up their HCV screening and treatment

programs. First, we found that in the general population, GT1 and GT3 account for a similar proportion of chronic HCV infected cases, followed closely by GT2. Therefore, pangenotypic treatment regimens recently introduced through the Georgian hepatitis C elimination program will likely have a positive impact on program performance and further increase treatment success rates [35, 36].

Second, our original analysis of the seroprevalence survey data (analyzing data from males and females together) found that approximately one-half of all seropositive respondents (38% of males and 70% of females) did not report either of the two risk factors associated with HCV infection (IDU or blood transfusion) [24]. The addition of the sex-specific explanatory models presented here identified two additional independent risk factors specific to females (history of surgery and dialysis) that reduced the number of seropositive females without a reported risk factor to 16%. However, due to the low number of HCV seropositive study participants reporting history of dialysis (n = 2), the effect estimate is imprecise (aOR = 7.3, 95% CI: 1.5, 35.3), and the strength of this association should be interpreted with caution. No additional behavioral or healthcare-related risk factors were identified in this analysis for males and 38% of seropositive males still did not have a reported risk factor, suggesting that screening programs cannot rely solely on targeting self-identified high-risk populations to identify HCV infections and eliminate hepatitis C. General population and/or age-targeted screening activities will also be necessary to achieve sufficient screening coverage to reach Georgia's hepatitis C elimination goal of identifying at least 90% of HCV-infected individuals by 2030 [27]. However, in addition to the screening programs in general population, maintaining the targeted interventions in the high-risk groups are necessary. For example, scaling up harm reduction services, hepatitis C testing and treatment for people who inject drugs will be essential. Georgia made substantial progress in this direction and initiated integration of hepatitis C treatment at harm reduction centers and among people receiving methadone substitution therapy [37, 38].

Third, in the absence of resources to support universal screening, such as in other low and middle-income countries with hepatitis C epidemiology similar to Georgia, an exposure index that incorporates demographic characteristics as well as behavioral risk factors associated with HCV infection could be used to target testing efforts. Screening based on exposure score could help prioritize testing efforts in groups most likely to be infected. However, our exposure index had good discriminatory performance only among males, leaving universal screening as the only option for identifying HCV-infected females who do not report risk factors. Further study is warranted to determine whether there are additional hepatitis C risk factors disproportionately affecting women, and also to test different approaches to asking women about their risk factor history to improve reporting (e.g. including questions about more detailed history of surgical procedures, such as C-section, in the survey questionnaire).

Our study has several limitations. First, the cross-sectional study design does not allow us to make causal claims regarding risk factors that may have occurred anytime during participants' lifetimes. Second, risk factor information was self-reported by participants and therefore subject to bias, including recall and social desirability biases. These potential biases may be one explanation for the 16% of female and 38% of male participants who did not report risk factors. Third, the study population did not include people experiencing incarceration or homelessness during the study period, groups both known to be at higher risk of HCV infection [39]. Therefore, our findings may not represent risk factors or genotype distribution in these subpopulations. Fourth, the exposure index we created would need to be externally validated using additional data sources if implemented in countries with different epidemiologic characteristics of HCV infection. Fifth, the small number of female participants reporting history of injection drug use, incarceration, tattoos, and kidney dialysis does not allow us to reliably estimate the HCV seroprevalence in women with the history of these risk factors.

In conclusion, our analysis confirms that to prevent further HCV transmission in Georgia, it is essential to scale up prevention interventions targeted to people who inject drugs and to improve quality control in blood donation services. In addition, the exposure index presented here may allow for further targeted testing that could increase efficiency and cost-effectiveness. The original analysis of these data demonstrated that, due to the high proportion of participants without reported risk factors, risk-based screening alone will not be sufficient to reach Georgia's goal to identify at least 90% of individuals with chronic HCV infection by 2030 [24]. The exposure index in this follow-on analysis offers a tool to help expand screening efforts to support hepatitis C elimination in Georgia and could be used in other countries with similar populations and risk factor profiles. However, to optimize testing, further studies may be needed to better understand potential exposures and/or to identify the most effective interview methods to improve disclosure of risk factors, particularly among females.

## Supporting information

**S1 Appendix.**
(DOCX)

## Acknowledgments

**Disclaimer**: The findings and conclusions in this report are those of the authors and do not necessarily reflect the official position of the Centers for Disease Control and Prevention, or the authors' affiliated institutions.

## Author Contributions

**Conceptualization:** Davit Baliashvili, Francisco Averhoff, Ana Kasradze, Giorgi Kuchukhidze, Amiran Gamkrelidze, Paata Imnadze, Gvantsa Chanturia, Nazibrola Chitadze, Curtis Blanton, Jan Drobeniuc, Juliette Morgan, Liesl M. Hagan.

**Data curation:** Davit Baliashvili, Ana Kasradze, Stephanie J. Salyer, Giorgi Kuchukhidze, Paata Imnadze, Maia Alkhazashvili, Gvantsa Chanturia, Nazibrola Chitadze, Roena Sukhiashvili, Curtis Blanton, Jan Drobeniuc, Juliette Morgan, Liesl M. Hagan.

**Formal analysis:** Davit Baliashvili, Curtis Blanton, Jan Drobeniuc, Liesl M. Hagan.

**Funding acquisition:** Francisco Averhoff, Stephanie J. Salyer, Amiran Gamkrelidze, Paata Imnadze, Juliette Morgan.

**Investigation:** Davit Baliashvili, Stephanie J. Salyer, Giorgi Kuchukhidze, Amiran Gamkrelidze, Paata Imnadze, Maia Alkhazashvili, Gvantsa Chanturia, Nazibrola Chitadze, Roena Sukhiashvili, Curtis Blanton, Jan Drobeniuc, Liesl M. Hagan.

**Methodology:** Davit Baliashvili, Francisco Averhoff, Ana Kasradze, Stephanie J. Salyer, Giorgi Kuchukhidze, Paata Imnadze, Maia Alkhazashvili, Gvantsa Chanturia, Nazibrola Chitadze, Roena Sukhiashvili, Curtis Blanton, Jan Drobeniuc, Juliette Morgan, Liesl M. Hagan.

**Project administration:** Francisco Averhoff, Ana Kasradze, Stephanie J. Salyer, Giorgi Kuchukhidze, Amiran Gamkrelidze, Paata Imnadze, Maia Alkhazashvili, Gvantsa Chanturia, Nazibrola Chitadze, Juliette Morgan, Liesl M. Hagan.

**Resources:** Francisco Averhoff, Ana Kasradze, Amiran Gamkrelidze, Maia Alkhazashvili, Gvantsa Chanturia, Nazibrola Chitadze, Roena Sukhiashvili, Jan Drobeniuc, Juliette Morgan, Liesl M. Hagan.

**Software:** Davit Baliashvili, Giorgi Kuchukhidze, Curtis Blanton, Liesl M. Hagan.

**Supervision:** Francisco Averhoff, Ana Kasradze, Stephanie J. Salyer, Giorgi Kuchukhidze, Amiran Gamkrelidze, Paata Imnadze, Maia Alkhazashvili, Gvantsa Chanturia, Nazibrola Chitadze, Roena Sukhiashvili, Jan Drobeniuc, Juliette Morgan, Liesl M. Hagan.

**Validation:** Davit Baliashvili, Maia Alkhazashvili, Gvantsa Chanturia, Nazibrola Chitadze, Roena Sukhiashvili, Jan Drobeniuc, Liesl M. Hagan.

**Visualization:** Davit Baliashvili, Liesl M. Hagan.

**Writing – original draft:** Davit Baliashvili, Liesl M. Hagan.

**Writing – review & editing:** Davit Baliashvili, Francisco Averhoff, Ana Kasradze, Stephanie J. Salyer, Giorgi Kuchukhidze, Amiran Gamkrelidze, Paata Imnadze, Maia Alkhazashvili, Gvantsa Chanturia, Roena Sukhiashvili, Curtis Blanton, Jan Drobeniuc, Juliette Morgan, Liesl M. Hagan.

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
