## [Decision Letter · Decision Letter 0]

5 Oct 2021

PONE-D-21-27334Risk factors and genotype distribution of hepatitis C virus in Georgia: a nationwide population-based surveyPLOS ONE

Dear Dr. Baliashvili,

Thank you for submitting your manuscript to PLOS ONE. After careful consideration, we feel that it has merit but does not fully meet PLOS ONE’s publication criteria as it currently stands. Therefore, we invite you to submit a revised version of the manuscript that addresses the points raised during the review process.

We look forward to receiving your revised manuscript.

Kind regards,

Jason T. Blackard, PhD

Academic Editor

PLOS ONE

Journal Requirements:

2. Please provide additional details regarding participant consent. In the ethics statement in the Methods and online submission information, please ensure that you have specified whether consent was written or verbal/oral. If consent was verbal/oral, please specify: 1) whether the ethics committee approved the verbal/oral consent procedure, 2) why written consent could not be obtained, and 3) how verbal/oral consent was recorded. If your study included minors, please state whether you obtained consent from parents or guardians in these cases. If the need for consent was waived by the ethics committee, please include this information.

 [This work was supported in part by the NIH Fogarty International Center Global Infectious Diseases grant D43TW007124 (to DB). The funders had no role in study design, data collection and analysis, decision to publish, or preparation of the manuscript.]

[This work was supported in part by the NIH Fogarty International Center Global Infectious Diseases grant D43TW007124 (to DB).]

 [This work was supported in part by the NIH Fogarty International Center Global Infectious Diseases grant D43TW007124 (to DB). The funders had no role in study design, data collection and analysis, decision to publish, or preparation of the manuscript.]

Additional Editor Comments:

This is a large cross-sectional study of HCV prevalence and genotypes in Georgia.  Given the movement towards a national elimination program, studies such as this are highly relevant.  The methods are well described, although several important details are missing and would strength the revised manuscript significantly.

HCV RNA was evaluated in samples but no information is given about the assay, genomic region targeted, or the lower limit of detection.

Was genotype evaluated for all HCV RNA positive samples?  It is unclear how many samples were HCV RNA positive but not tested for genotype or gave a genotype PCR negative result.

What region was used for HCV genotyping?  The PCR primers and the genomic region amplified must be provided.

Where is the phylogenetic tree for HCV genotyping?  This is an important element of such studies.

Lines 213-214:   how were samples assigned to the model building analysis versus the validation group?  If not randomly, then how?

Line 226:  if no variables were identified using a 60% subset of the larger dataset, then why wasn’t 100% of the dataset used?

Line 251:  how is the universal screening done?  Antibody only or nucleic acid testing?

The lack of identifying IDU or blood transfusion in many individuals is quite disturbing.  If this is stigmatizing, then the study instrument / data collection approach is inadequate.  Self-report is not the only way to collect these data!

Reviewers' comments:

Reviewer's Responses to Questions

**Comments to the Author**

1. Is the manuscript technically sound, and do the data support the conclusions?

Reviewer #1: Yes

Reviewer #2: Partly

2. Has the statistical analysis been performed appropriately and rigorously? 

Reviewer #1: Yes

Reviewer #2: Yes

3. Have the authors made all data underlying the findings in their manuscript fully available?

Reviewer #1: No

Reviewer #2: Yes

4. Is the manuscript presented in an intelligible fashion and written in standard English?

Reviewer #1: Yes

Reviewer #2: Yes

5. Review Comments to the Author

Reviewer #1: The study reported HCV genotype distribution and potential risk factors contributing to HCV transmission in Georgia. This study provides useful information and contributes substantially to the epidemiology of HCV in preparation of the elimination in the country. While publication is recommended, the manuscript requires minor revision.

Laboratory methodology

The author should elaborate more on the method used for genotyping. The previous article they are referring too does not mention how HCV genotype was obtained.

Results

Line 183: These results do not correlate to Figure 2. “Among anti-HCV positive males, 50.9% reported history of IDU, and 16.9% reported history of blood transfusion; among anti-HCV positive females, 2.4% reported history of IDU, 27.6% reported receiving a blood transfusion, and 80% reported history of surgery”; while figure 2 indicate different results: that among anti-HCV positive males, 44.7% reported history of IDU, and 10.7% reported history of blood transfusion; among anti-HCV positive females, 2.0% reported history of IDU, 2.3% reported receiving a blood transfusion, and 54.4% reported history of surgery.

Line 202-203: GT1b was most prevalent among females (61.5%) and…..These results do not correlate to Table 2; Table 2 report 62.5%

Table 2, overall percentages are less than 100% for sex (males) and age groups 18-29, 40-49, and 50-59

Discussion

Line 243: In the discussion, the authors state that GT1b was more common among females, persons more likely infected via blood transfusion and persons over the age of 50, while GT3 was more common among males, persons more likely infected through IDU, and younger age groups and they further state that pangenotypic program will likely have a positive impact on program performance and further increase treatment success rates. However, they did not support their study in comparison with other studies done in the country. Which other common genotypes did other studies report?

Line 256: History of surgery was reported by a larger proportion of seropositive females (80%) than males (53%) and was associated with anti-HCV positivity only among females. As indicated above in the results section these results do not correlate with Figure 2. The authors should revise this statement based on the correct results and correct figure.

The discussion section should be reorganized

Reviewer #2: Major

1. In this manuscript, the authors used the cohort of the epidemiological studies reported in 2019 (ref. 24) [Hagan LM, et al. Hepatitis C prevalence and risk factors in Georgia, 2015: setting a baseline for elimination, BMC Public Health. 2019;19(3):480], and re-analyzed by dividing with gender. However, the predictive model for females has a low AUC of 0.61 (page3 line46), and its usefulness as a predictive model is limited.

2. Larger number of women with a history of surgery in HCV-infected patients are enrolled, however, there is no comments with the type of surgery, nor comments with the blood transfusion during the surgery.

3. The differences from previous reports, or the characteristics of the state of Georgia in US were not described in the manuscript.

4. In Table 2, the “reported risk factor” and “age groups” should also be divided by gender like “people with no reported risk factors”.

5. In Table 4, the anti-HCV positivity is identified by the exposure index score. However, there is no description about the threshold score for the definition positive. Moreover, the authors need to show the characteristic clinical feature in patients with high score.

6. In the figure S1, the number of the group “blood transfusion only” was decreased in the age of 30-19 years old. Why? In addition, the reviewer cannot understand the reason why the number of the group “neither” was increased in the elder patients’ group.

7. The risk factors of HCV-infected patients are discussed, however, the information how to use the results of the study to enclose the patients, and link them to the anti-viral treatment in clinics.

Minor

1. in the figure legend of figure S3, the description of "b" is lacking.

2. There is no figure legends in the figure S1 and S2.

6. PLOS authors have the option to publish the peer review history of their article (what does this mean?). If published, this will include your full peer review and any attached files.

Reviewer #1: No

Reviewer #2: No

---

## [Author Response · Author response to Decision Letter 0]

9 Dec 2021

Additional Editor Comments:

This is a large cross-sectional study of HCV prevalence and genotypes in Georgia. Given the movement towards a national elimination program, studies such as this are highly relevant. The methods are well described, although several important details are missing and would strength the revised manuscript significantly.

Comment 1: HCV RNA was evaluated in samples but no information is given about the assay, genomic region targeted, or the lower limit of detection.

Response 1: We expanded the laboratory methodology subsection of the methods section to include these details (lines 96-100). Briefly, genotyping was performed using commercial kit - HCV Real-TM Genotype from Sacace. This Real Time PCR Kit was dedicated for qualitative detection and differentiation of hepatitis C virus (HCV) genotypes 1a, 1b, 2, 3, 4. The manufacturer does not specify the target regions. Analytical sensitivity provided in the instruction is 500 IU/ml. 

Comment 2: Was genotype evaluated for all HCV RNA positive samples? It is unclear how many samples were HCV RNA positive but not tested for genotype or gave a genotype PCR negative result.

Response 2: All RNA-positive samples identified by the Sacace assay were processed for genotyping (n=310). One additional sample was tested positive for RNA after retesting the samples at the CDC, hence the previous paper reports 311 participants with RNA positive results. However, the genotyping results from that additional sample was not available in our data and our sample size for genotyping analysis was 310. We added the information on RNA positivity by sex in the results section (lines 164-166). Five participants with indeterminate genotype results, mainly with high CT, were removed from further analysis and distribution of genotypes by gender, age and risk factors includes 305 individuals, as described in the table 2. 

Comment 3: What region was used for HCV genotyping? The PCR primers and the genomic region amplified must be provided.

Response 3: Unfortunately manufacturer does not specify the target region and the primer/probe information.

Comment 4: Where is the phylogenetic tree for HCV genotyping? This is an important element of such studies.

Response 4: We agree with the editor that phylogenetic tree is an important element of such studies. However, since the Sacase assay used in this study is a commercial HCV genotyping test, phylogenetic tree for HCV genotyping was not required. It would have been required if the HCV genotyping was determined by sequencing HCV+ samples, but that was not the case in our study. 

Comment 5: Lines 213-214: how were samples assigned to the model building analysis versus the validation group? If not randomly, then how?

Response 5: Samples were randomly assigned to model building versus validation group. We added this detail in the text and the updated sentence reads as follows: "The model was built using randomly selected 60% of the male subset of the serosurvey data (n=1,490) and validated on the remaining 40% (n=938)" (line 220).

Comment 6: Line 226: if no variables were identified using a 60% subset of the larger dataset, then why wasn’t 100% of the dataset used?

Response 6: Since we did not have any external data to validate the model, we decided to use splitting the data into training and validation data sets. Using 100% of the data in our predictive model would not be feasible and even if we were able to identify any significant variables, it would not be appropriate to report them as significant without validation set.

Comment 7: Line 251: how is the universal screening done? Antibody only or nucleic acid testing?

Response 7: Universal screening of blood donations was conducted using antibody testing. We added this detail in the line 258.

Comment 8: The lack of identifying IDU or blood transfusion in many individuals is quite disturbing. If this is stigmatizing, then the study instrument / data collection approach is inadequate. Self-report is not the only way to collect these data!

Response 8: We agree with the editor that the lack of self-reported major risk factors in large proportion of participants is less than ideal. This has been acknowledged in the original manuscript and was taken into account in the programmatic planning of Georgian Hepatitis C elimination program by emphasizing the screening in general population rather than just high-risk groups. Unfortunately, during the implementation of this seroprevalence survey, it was not feasible to obtain the information on these risk factors from other sources besides self-report. We describe this limitation in the discussion section (lines 330-32). We tried to address this issue by conducting analysis stratified by sex presented in this manuscript and identified history of surgery as an additional risk factor among females. 

Reviewers' comments:

Laboratory methodology

Comment 1: The author should elaborate more on the method used for genotyping. The previous article they are referring too does not mention how HCV genotype was obtained.

Response 1: We agree with the reviewer that more details are necessary in the description of the genotyping. We expanded the laboratory methodology subsection in Methods to include additional information, such as manufacturer and analytical sensitivity (lines 96-100).

Results

Comment 2: Line 183: These results do not correlate to Figure 2. “Among anti-HCV positive males, 50.9% reported history of IDU, and 16.9% reported history of blood transfusion; among anti-HCV positive females, 2.4% reported history of IDU, 27.6% reported receiving a blood transfusion, and 80% reported history of surgery”; while figure 2 indicate different results: that among anti-HCV positive males, 44.7% reported history of IDU, and 10.7% reported history of blood transfusion; among anti-HCV positive females, 2.0% reported history of IDU, 2.3% reported receiving a blood transfusion, and 54.4% reported history of surgery.

Response 2: We appreciate the reviewers feedback regarding the text/figure discrepancy. The differences are caused by the fact that the percentages in figure is presented separately for participants with single risk factors and those with multiple factors, while the text combines them together. For example, 50.9% of males with reported history of IDU mentioned in the text is a sum of 44.7% of males who reported only IDU and 6.2% of males who reported both IDU and blood transfusion. To avoid confusion in the reader, we updated the text, and the revised sentences reads as follows: “Among anti-HCV positive males, 50.9% reported history of IDU, and 16.9% reported history of blood transfusion, including 6.2% who reported both of those risk factors; among anti-HCV positive females, 2.4% reported history of IDU, 27.6% reported receiving a blood transfusion, and 80% reported history of surgery, including 25.7% who reported a combination of these factors” (lines 189-191).

Comment 3: Line 202-203: GT1b was most prevalent among females (61.5%) and…..These results do not correlate to Table 2; Table 2 report 62.5%

Response 3: We thank reviewer for noticing the discrepancy between table and text. There was a typo in the text, and we changed the number 61.5% to 62.5% (line 210).

Comment 4: Table 2, overall percentages are less than 100% for sex (males) and age groups 18-29, 40-49, and 50-59

Response 4: Overall percentages within the categories mentioned by the reviewer sum up to 99.9% due to the rounding error. We decided to keep the percentages rounded to one decimal point to make sure the tables are concise and easy to read, but we added a note to the table mentioning that percentages within each category might not sum up to 100% due to the rounding error. 

Discussion

Comment 5: Line 243: In the discussion, the authors state that GT1b was more common among females, persons more likely infected via blood transfusion and persons over the age of 50, while GT3 was more common among males, persons more likely infected through IDU, and younger age groups and they further state that pangenotypic program will likely have a positive impact on program performance and further increase treatment success rates. However, they did not support their study in comparison with other studies done in the country. Which other common genotypes did other studies report?

Response 5: We agree with the reviewer about importance to comparing the findings to other studies in the country. This was the first nationwide survey studying the prevalence and genotype distribution of HCV infection the in Georgia, which unfortunately limits our ability to compare our findings to previous reports and note any differences. We added a paragraph in the discussion where we compare our findings with the previously study in the capital city Tbilisi (lines 285-292), but we also note our limited ability to make direct comparisons considering the different populations included in the two studies (The whole country in our study vs. capital city in the previous study).

Comment 6: Line 256: History of surgery was reported by a larger proportion of seropositive females (80%) than males (53%) and was associated with anti-HCV positivity only among females. As indicated above in the results section these results do not correlate with Figure 2. The authors should revise this statement based on the correct results and correct figure.

Response 6: As responded above in comment 2, the differences are caused by the fact that the percentages in figure is presented separately for those with single risk factors and those multiple factors. To avoid confusion, we added clarification in both, results and discussion section. (lines 189-191 and 263).

Comment 7: The discussion section should be reorganized

Response 7: We added a new paragraph in the discussion section discussing the comparison of our finding with the previous study (lines 285-292) and expanded the paragraph about the implications of our findings for testing and treatment (lines 312-317). We will be happy to make additional changes in the discussion if reviewer has any specific suggestions. 

Reviewer #2: Major

Comment 1. In this manuscript, the authors used the cohort of the epidemiological studies reported in 2019 (ref. 24) [Hagan LM, et al. Hepatitis C prevalence and risk factors in Georgia, 2015: setting a baseline for elimination, BMC Public Health. 2019;19(3):480], and re-analyzed by dividing with gender. However, the predictive model for females has a low AUC of 0.61 (page3 line46), and its usefulness as a predictive model is limited.

Response 1: We agree with the reviewer that the predictive model for females has limited usefulness. Therefore, we are not suggesting its use in practice, and we did not use it to create an exposure index for females.

Comment 2: Larger number of women with a history of surgery in HCV-infected patients are enrolled, however, there is no comments with the type of surgery, nor comments with the blood transfusion during the surgery.

Response 2: Unfortunately the survey instrument did not include more details about the type of surgery or blood transfusion during the surgery. 

Comment 3: The differences from previous reports, or the characteristics of the state of Georgia in US were not described in the manuscript.

Response 3: We agree with the reviewer about importance to comparing the findings to other studies in the country. This was the first nationwide survey studying the prevalence and genotype distribution of HCV infection the in Georgia, which unfortunately limits our ability to compare our findings to previous reports and note any differences. We added a paragraph in the discussion section to describe a previous study conducted in the capital city of Georgia and compared its findings to the results from our study (285-292).

To clarify, the study was conducted in the Eastern European country of Georgia, not US state of Georgia. 

Comment 4: In Table 2, the “reported risk factor” and “age groups” should also be divided by gender like “people with no reported risk factors”.

Response 4: We agree with the reviewer about the need to provide gender-specific genotype distribution. We reorganized and expanded the table 2 accordingly. 

Comment 5: In Table 4, the anti-HCV positivity is identified by the exposure index score. However, there is no description about the threshold score for the definition positive. Moreover, the authors need to show the characteristic clinical feature in patients with high score.

Response 5: Authors would like to clarify that the anti-HCV positivity was not identified by the exposure index score. Table 4 shows the actual anti-HCV positivity determined by the laboratory testing of blood samples, therefore, it did not require using any threshold. We realize that the title of the table might have been misleading. To avoid the confusion, we changed the title, which now reads as follows: Proportions of anti-HCV positive participants in each of the exposure score categories among males, Georgia HCV serosurvey, 2015.

Comment 6: In the figure S1, the number of the group “blood transfusion only” was decreased in the age of 30-19 years old. Why? In addition, the reviewer cannot understand the reason why the number of the group “neither” was increased in the elder patients’ group.

Response 6: The numbers provided on figure S1 represent the percentages of the participants in respective age groups that reported the given risk factor. To provide clearer explanation of the figure, we added a note in the legend of the figure. The proportion of the participants who report neither risk factors was higher in older age groups, which could be explained by the recall bias. The authors discuss the implications of this finding in the fourth paragraph of the discussion section (lines 279-281).

Comment 7: The risk factors of HCV-infected patients are discussed, however, the information how to use the results of the study to enclose the patients, and link them to the anti-viral treatment in clinics.

Response 7: We agree with the reviewer about the importance to include more information about the implications of the study results. We expanded one of the paragraph in the discussion section where we describe the implications of our findings, and added a discussion about the importance of maintaining services targeted at high risk groups, mainly people who inject drugs (lines 312-317). 

Minor

Comment 1: in the figure legend of figure S3, the description of "b" is lacking.

Response 1: The description in the footnote “a” is applicable to both panels a and b. To avoid confusion, we removed the superscripted “a” from the legend and replaced it with “Note”. 

Comment 2: There is no figure legends in the figure S1 and S2.

Response 2: We appreciate the reviewer noting the missing legends. We added legends to describe the information provided on the figures S1 and S2.

---

## [Decision Letter · Decision Letter 1]

10 Jan 2022

Risk factors and genotype distribution of hepatitis C virus in Georgia: a nationwide population-based survey

PONE-D-21-27334R1

Dear Dr. Baliashvili,

We’re pleased to inform you that your manuscript has been judged scientifically suitable for publication and will be formally accepted for publication once it meets all outstanding technical requirements.

Kind regards,

Jason T. Blackard, PhD

Academic Editor

PLOS ONE

Additional Editor Comments (optional):

None

Reviewers' comments:

Reviewer's Responses to Questions

**Comments to the Author**

1. If the authors have adequately addressed your comments raised in a previous round of review and you feel that this manuscript is now acceptable for publication, you may indicate that here to bypass the “Comments to the Author” section, enter your conflict of interest statement in the “Confidential to Editor” section, and submit your "Accept" recommendation.

Reviewer #1: All comments have been addressed

Reviewer #2: All comments have been addressed

2. Is the manuscript technically sound, and do the data support the conclusions?

Reviewer #1: Yes

Reviewer #2: Partly

3. Has the statistical analysis been performed appropriately and rigorously? 

Reviewer #1: Yes

Reviewer #2: Yes

4. Have the authors made all data underlying the findings in their manuscript fully available?

Reviewer #1: Yes

Reviewer #2: Yes

5. Is the manuscript presented in an intelligible fashion and written in standard English?

Reviewer #1: Yes

Reviewer #2: Yes

6. Review Comments to the Author

Reviewer #1: The authors have addressed all the minor comments requested to my satisfaction. It is acceptable for publication.

Reviewer #2: I can accept for the authors' comments. I can understand the limitation of this study, and agreed with the comments that I think the authors have made efforts as much as they can.

7. PLOS authors have the option to publish the peer review history of their article (what does this mean?). If published, this will include your full peer review and any attached files.

Reviewer #1: **Yes: **Maemu Gededzha

Reviewer #2: No

---

## [Editor Report · Acceptance letter]

12 Jan 2022

PONE-D-21-27334R1 

Risk factors and genotype distribution of hepatitis C virus in Georgia: a nationwide population-based survey 

Dear Dr. Baliashvili:

I'm pleased to inform you that your manuscript has been deemed suitable for publication in PLOS ONE. Congratulations! Your manuscript is now with our production department. 

Kind regards, 

on behalf of

Dr. Jason T. Blackard 

Academic Editor

PLOS ONE